# DOCSPLIT: Simple Contrastive Pretraining for Large Document Embeddings

**Yujie Wang**
Claremont Graduate University
`yujie.wang@cgu.edu`

**Mike Izbicki**
Claremont McKenna College
`Michael.Izbicki@ClaremontMcKenna.edu`

## Abstract

Existing model pretraining methods only consider local information. For example, in the popular token masking strategy, the words closer to the masked token are more important for prediction than words far away. This results in pretrained models that generate high-quality sentence embeddings, but low-quality embeddings for large documents. We propose a new pretraining method called DOCSPLIT which forces models to consider the entire global context of a large document. Our method uses a contrastive loss where the positive examples are randomly sampled sections of the input document, and negative examples are randomly sampled sections of unrelated documents. Like previous pretraining methods, DOCSPLIT is fully unsupervised, easy to implement, and can be used to pretrain any model architecture. Our experiments show that DOCSPLIT outperforms other pretraining methods for document classification, few shot learning, and document retrieval tasks.

## 1 Introduction

Generating high-quality text embeddings for documents is a long-standing open problem. Most previous studies focus on either learning sentence-level representations (Hill et al., 2016; Logeswaran and Lee, 2018; Gao et al., 2021) where training data usually contain short text or designing specific model structures for larger-range dependencies (Beltagy et al., 2020; Zaheer et al., 2020), but effective and efficient document representation learning methods are less explored.

This paper presents DOCSPLIT, the first unsupervised pretraining method designed specifically for large documents. DOCSPLIT is simple to use, and it can be applied to any model architecture to improve document representations. DOCSPLIT uses contrastive learning, and our key contribution is a new method for generating positive samples for contrastive learning. Figure 1 provides a graphical

Figure 1: In DOCSPLIT pretraining, an input document ($x$) is split into two new document summaries ($x_1^+$ and $x_2^+$) by randomly assigning each sentence to one of the new summaries. These new summaries are then used as positive instances for contrastive learning, which forces the model to represent these summaries with similar embeddings.

illustration of the method, and Section 2 describes the implementation details and intuition.

Section 3 describes how DOCSPLIT improves prior work. We begin by demonstrating that prior work on contrastive pretraining has focused only on sentence-level representations. Then we show that work on model architectures designed for large documents has ignored the problem of better pretraining methods designed to work with these architectures.

In Section 4, we describe two pretrained models based on the BERT architecture (for fair comparison to other contrastive losses) and the LongFormer architecture (for optimal performance on large documents). We evaluate these models on standard large document classification, few shot learning, and document retrieval tasks. We find that models pretrained using DOCSPLIT significantly outperform models pretrained with all other published methods.

## 2 Method

We first review contrastive learning at a high level, then we describe our DOCSPLIT method.

### 2.1 Contrastive Learning

Contrastive Learning learns effective representations by pulling semantically close neighbors together and pushing apart non-neighbors in the latent space (Hadsell et al., 2006). Each data point $x$ is converted into a contrastive instance $\{x_1^+, x_2^+, x_1^-, \ldots, x_N^-\}$ that includes two positive examples and $N$ negative examples. Intuitively, the positive instances should be semantically similar to each other (they will be pushed together) and the negative instances should be semantically dissimilar to the positive instances (they will be pushed away). Following recent influential work (Chen et al., 2020; Gao et al., 2021; Li et al., 2022), we use cross entropy as our loss function. As the mathematical details are technical and not relevant to our discussion, we do not reproduce them here.

There are many model training methods that use contrastive learning, and what differentiates these methods is how they convert the data point $x$ into the contrastive instance. It is standard practice to use in-batch instances for the negative pairs, and so all that remains is designing a method for constructing the positive pairs $(x_1^+, x_2^+)$.

### 2.2 Document Splitting (DOCSPLIT)

Our main contribution is the DOCSPLIT method to construct the positive pairs needed for contrastive learning. The idea is simple to describe and implement. Given an input document, we first split the document into sentences. Then each sentence is randomly assigned to either the $x_1^+$ or $x_2^+$ document.

This splitting procedure results in two documents that can be thought of as "summaries" of the original document. These summaries will have similar semantic content, and the contrastive loss will ensure that these two documents have similar embeddings. Notice that because there is randomness in the document splitting procedure, multiple passes over the same data points will result in different contrastive instance pairs being generated. This provides built-in overfitting resistance through data augmentation when multiple epochs of the training data are used.

## 3 Related Work

There are two categories of related work: models trained using contrastive learning, and models designed for large documents.

### 3.1 Contrastive Pretraining

Contrastive learning has shown remarkable recent success for developing sentence embeddings. The simplest method is SimCSE (Gao et al., 2021), which uses dropout to generate correlated positive samples. The contrastive tension method (Carlsson et al., 2021) is similarly generic, but has a much more complicated implementation involving multiple models trained jointly. Because both of these pretraining strategies are generic, they can be used with any type of input including documents; but they do not take explicit advantage of document structure. The INSTRUCTOR model (Su et al., 2022) is the current state-of-the-art model for most downstream tasks. The contrastive objective for this model requires a specially constructed corpus of manual-human annotations, and this corpus is limited only to sentence-level annotations instead of document-level annotations. We show that our model significantly improves on INSTRUCTOR on document level tasks, and it is not clear how to extend the INSTRUCTOR model to document-level tasks because human annotation for documents is significantly more expensive than for sentences.

The Contriever model (Izacard et al., 2021) uses a contrastive objective most similar to our own. They use the document cropping and inverse cloze tasks for pretraining. In document cropping, a document is divided in half and the two halves are used as the positive samples; in inverse cloze, a contiguous substring of the document is used as one positive sample and all other strings are used as the negative sample. The DOCSPLIT pretraining method can be seen as a generalization of these methods.

### 3.2 Large Document Architectures

All of the models discussed in Section 3.1 above are based off of the BERT architecture (Devlin et al., 2019). This architecture uses an attention mechanism that requires $O(n^2)$ memory and runtime, where $n$ is the size of the attention window. The maximum size of a document that a model can understand is limited by this window size, and so compute for these models scales quadratically with the length of the documents.

A growing body of research focuses on developing new architectures with reduced computational requirements that enable processing larger documents. The LongFormer (Beltagy et al., 2020) and BigBird (Zaheer et al., 2020) models pioneered this line of research, and both models reduce the runtime of the attention mechanism to $O(n)$. A variety of other architectures have subsequently been proposed (e.g. **?????**). **?** provide a survey of this large body of work. Importantly, all of this research focuses only on improving the computational aspects of model architecture, and none of these models use a training objective designed specifically for large documents. Because the DOCSPLIT pretraining method is model agnostic, we can easily apply it to any of these newly proposed model architectures. For computational reasons, we limit our experimental comparisons in Section 4 below to the LongFormer and BigBird models since these are the two most influential model architectures designed for large documents. We find a large performance improvement when these models are trained with DOCSPLIT, and expect this performance improvement would extend to similar models as well.

## 4 Experiments

We perform a careful ablation study to isolate the effects of the DOCSPLIT pretraining method on downstream task performance. First, we pretrain separate models for each group of baseline models described in Section 3 above. Then, we perform downstream experiments on standard classification, few-shot learning, and document retrieval tasks. In all cases, our pretrained models significantly outperform prior work.

### 4.1 Pretraining Details

We pretrain two models on two different architectures. All prior work using contrastive learning discussed in Section 3.1 above evaluates their pretraining methods on the BERT architecture. To fairly compare against these methods, we pretrain our DOCSPLIT_bert model also on this architecture. Ultimately, however, we are interested in large document performance, and so we expect that a model architecture designed specifically for large documents will improve performance. We therefore also pretrain the DOCSPLIT_long model on the LongFormer architecture. This second model will be used to evaluate against other models designed specifically for large documents.

To pretrain both models, we follow the standard pretraining procedure for contrastive losses established by Gao et al. (2021) and Li et al. (2022). We simultaneously optimize both the masked language model (MLM) loss (with weight= 0.1) and the contrastive loss (with temperature $\tau = 0.05$). We use English Wikipedia articles as our pretraining dataset. These articles are long, and so we expect that a pretraining procedure designed for large documents will improve performance. The total number of training instances is 6,218,825. We use AdamW (Kingma and Ba, 2014) with a learning rate of 5e-5. DOCSPLIT_bert uses a batch size of 36. And due to the larger memory requirements of the LongFormer architecture, DOCSPLIT_long uses a batchsize of 12. For both models, we know that performance improvements on downstream tasks must be due to the pretraining procedure and not the dataset because all baseline models include English language wikipedia in their training set.

### 4.2 Experiment 1: Text Classification

We fine tune DOCSPLIT_bert, DOCSPLIT_long, and all baseline models on five standard document datasets. The datasets are summarized in the table below:[1]

| Dataset | Num Docs | Classes | Words / Doc | |
|---------|----------|---------|------|-----|
| | | | Mean | Max |
| FakeNews | 8,558,957 | 15 | 467 | 33,936 |
| arXiv | 2,162,833 | 38 | 138 | 925 |
| 20News | 18,846 | 20 | 258 | 11,554 |
| NYT | 13,081 | 5 | 650 | 5,503 |
| BBCNews | 2,225 | 5 | 133 | 445 |

Table 1 shows the accuracy and F1 score of every model on these datasets. Notice that DOCSPLIT_bert out performs all BERT-based models discussed in Section 3.1, and DOCSPLIT_long outperforms all models on every dataset. There are no results for INSTRUCTOR on the 20News dataset because INSTRUCTOR was pretrained on 20News, and the authors state that evaluating INSTRUCTOR on datasets it was pretrained is incorrect due to data contamination.

---

[1]Citations for the datasets are: Fake News Corpus https://github.com/several27/FakeNewsCorpus; arXiv articles dataset https://www.kaggle.com/datasets/Cornell-University/arxiv; 20NewsGroups (Lang, 1995); New York Times Annotated Corpus (NYT) (Sandhaus, 2008); and BBC-News http://mlg.ucd.ie/datasets/bbc.html.

Table 1: In the text classification of Experiment 1, models pretrained with DOCSPLIT outperform baseline models in all cases. Larger numbers are better.

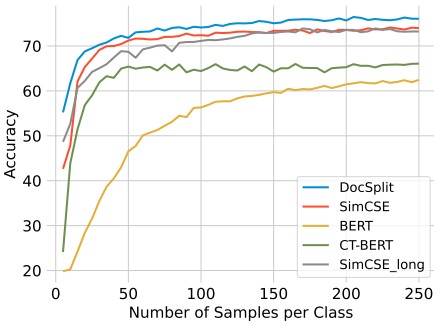

Figure 2: DOCSPLIT$_{\text{bert}}$ outperforms all other BERT-based models in a few-shot classification task on the 20News dataset.

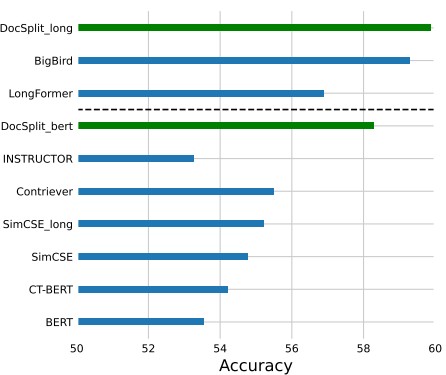

Figure 3: In the standard LRA document retrieval task, DOCSPLIT$_{\text{bert}}$ outperforms all other BERT-based models, and DOCSPLIT$_{\text{long}}$ outperforms all other models.

## 4.3 Experiment 2: Few-shot Learning

Next we evaluate how DOCSPLIT pretraining performs on classification tasks with a small number of training examples. We follow the standard procedure of artificially limiting the number of training examples used during training, and evaluating on the same test set. Figure 2 shows the classification accuracy on the 20News dataset as we vary the size of the training set. We see that DOCSPLIT$_{\text{bert}}$ outperforms all other BERT-based models accross all sample sizes.

The results on other datasets and for LongFormer-based models are similar. A full set of results on other datasets is available in the Appendix.

## 4.4 Experiment 3: Document Retrieval

We follow the document retrieval experiment of the Long Range Arena (LRA) benchmark (Tay et al., 2020), which uses the ACL Anthology Network (AAN) dataset (Radev et al., 2009). The goal of the task is to predict whether two papers have a citation link given only their embeddings, which is a common setup used in long-form document matching (Jiang et al., 2019; Yang et al., 2020). This reduces to a binary classification task and accuracy is the commonly reported performance measure. Figure 3 shows the results. DOCSPLIT$_{\text{bert}}$ outperforms all other BERT-based models, and DOCSPLIT$_{\text{long}}$ outperforms all other models.

## 5 Conclusion

DOCSPLIT is the first unsupervised pretraining method designed explicitly for large documents. DOCSPLIT can be used to pretrain any model architecture, and we provide code to do so at

```
https://blinded_for_review
```

The LongFormer-based DOCSPLIT$_{\text{long}}$ model provides SOTA performance on document classification, few-shot learning, and retrieval tasks.

## Limitations

We identify three limitations with our work.

**Limitation 1:** We evaluate DOCSPLIT using only three downstream tasks (classification, few-shot learning, and document retrieval), but pretrained models are useful in many other tasks as well. For example, the LongFormer (Beltagy et al., 2020) is evaluated also on question answering and coreference resolution. We did not run these experiments only due to a lack of computational resources, and we hope that future work will evaluate the performance of DOCSPLIT pretraining on other downstream tasks as well. We note, however, that our DOCSPLIT experiments in some ways are more extensive than other papers' experiments. For example, the LongFormer paper does not include any document retrieval experiments, and we do. Also, the Contriever model (Izacard et al., 2021) was specifically designed for pretraining models for the downstream retrieval experiment task and only evaluates on this downstream task. We beat

the Contriever model on this task and provide the classification and few-shot experiments in addition.

Much of the prior work on contrastive learning evaluates on a large number of tasks as well. For example, INSTRUCTOR (Su et al., 2022) is evaluated on 70 different downstream tasks. None of these tasks, however, are applicable in our setting because these tasks include datasets with only small text and not large documents.

**Limitation 2:** Our models are pretrained only on English wikipedia, but other models are pretrained on significantly larger datasets. For example, the LongFormer model was pretrained on English wikipedia and on the Books (**?**), Real News (**?**), and Stories (**?**) datasets. We pretrained on this smaller dataset only due to our limited computational resources. Results in scaling laws (**?**) suggest that pretraining on more data would significantly improve our models' performance, and so our results are reported results are likely underestimating the positive effects of DOCSPLIT pretraining.

**Limitation 3:** Our experiments do not evaluate how well DOCSPLIT will work on extremely large (e.g. book length) documents. Existing large document model architectures like LongFormer (Beltagy et al., 2020) and BigBird (Zaheer et al., 2020) are still not efficient enough to train models on these extremely large documents. More work also needs to be done on generating suitable evaluation tasks in this extreme setting to understand DOCSPLIT performance.

## Ethics Statement

Learning embeddings is a standard problem in natural language processing. Our approach uses standard training datasets and training procedures. There are therefore no direct ethical concerns with this research.

Our total compute is relatively small and so we have a small environmental impact. Pretraining DOCSPLIT$_{bert}$ took 4 days on an Nvidia 2080 GPU with 11GB of RAM, and pretraining DOCSPLIT$_{long}$ took 4 days with an Nvidia Quadro RTX 8000 with 48GB of RAM. Other models use larger GPU clusters that require considerably more energy. The finetuning procedures run in less than 1 day on the same systems.

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

| Datasets | FakeNews | | 20News | | arXiv | | NYT | | BBCNews | |
|---|---|---|---|---|---|---|---|---|---|---|
| Metrics | Acc | F1 | Acc | F1 | Acc | F1 | Acc | F1 | Acc | F1 |
| *Few-shot Text Classification* | | | | | | | | | | |
| BERT | 23.96 | 23.73 | 19.94 | 18.71 | 24.08 | 10.14 | 51.85 | 43.90 | 54.22 | 52.73 |
| CT-BERT | 23.71 | 23.06 | 24.11 | 23.53 | 27.02 | 13.53 | 47.23 | 36.83 | 59.56 | 58.95 |
| SimCSE | 25.04 | 22.68 | 42.63 | 41.42 | 32.61 | 17.19 | 86.51 | 78.41 | 83.56 | 83.75 |
| SimCSE$_{long}$ | 26.39 | 23.26 | 48.65 | 47.81 | 23.42 | 12.66 | 85.36 | 75.90 | 84.44 | 83.96 |
| Contriever | 24.17 | 22.35 | 54.27 | 54.87 | 33.19 | 17.23 | 84.73 | 77.62 | 77.71 | 79.19 |
| INSTRUCTOR | 26.11 | 23.97 | – | – | 33.51 | 17.12 | 32.38 | 27.60 | 52.67 | 48.78 |
| DOCSPLIT$_{bert}$ | **27.79** | **24.65** | **55.79** | **55.43** | **35.79** | **18.52** | **90.52** | **83.71** | **86.86** | **86.31** |
| LongFormer | 26.56 | 25.12 | 44.42 | 42.41 | 25.04 | 13.36 | 73.06 | 54.87 | 84.89 | 85.47 |
| BigBird | 25.36 | 23.28 | 39.14 | 39.06 | 23.62 | 10.18 | 86.66 | 78.96 | 79.11 | 76.63 |
| DOCSPLIT$_{long}$ | **29.17** | **27.13** | **51.18** | **50.96** | **34.33** | **18.80** | **89.78** | **82.88** | **86.78** | **86.66** |

Table 2: Few-shot classification on five datasets.

# A Appendix

## A.1 Training Details

For text classification, the learning rate for fine-tuning is 3e-4; the batch size is 8; the maximum sequence length is 512 tokens. We fine-tune the last MLP layer on these five datasets and evaluate the classification performance with accuracy and macro-F1 scores. For few-shot text classification, we sample 10 data instances per class for the FakeNewsCorpus dataset and the arXiv dataset and 5 data instances per class for the other three datasets. Other settings are the same as the standard text classification. Since there is randomness in sampling, we repeat every experiment 10 times and take the average value of metrics.

## A.2 Few-shot Learning

Table 2 shows the results of few-shot text classification on these five datasets. We can see that, under the same model structure, DOCSPLIT (i.e., DOCSPLIT$_{bert}$ and DOCSPLIT$_{long}$) achieves 12.0% and 24.3% macro-F1 improvements compared to SimCSE and Longformer respectively. Surprisingly, INSTRUCTOR achieves low performance on the NYT and BBCNews under few-shot settings. These improvements are higher than standard text classification.