# OpenReview forum: "DocSplit: Simple Contrastive Pretraining for Large Document Embeddings"
_EMNLP/2023/Conference — EMNLP 2023 Findings_

### Official Review · Reviewer_c7xp · 2023-08-02

**Soundness:** 4

**Excitement:**

3: Ambivalent: It has merits (e.g., it reports state-of-the-art results, the idea is nice), but there are key weaknesses (e.g., it describes incremental work), and it can significantly benefit from another round of revision. However, I won't object to accepting it if my co-reviewers champion it.

**Paper Topic And Main Contributions:**

Existing model pretraining methods do not work well in generating embeddings for large documents because they consider local information. The authors propose a novel method -- DocSplit -- which forces the model to consider more global information through contrastive methods. Experimental results demonstrate the efficacy of the proposed methods.

**Reasons To Accept:**

- Well-motivated problems and clear writing.
- Well-executed experiments.
- Simple methods that are very practical. Also, this is not necessarily a reason for acceptance, I appreciate that the authors chose not to include (seemingly unnecessary) technical details for clarity (in Lines 079-081).
- Somewhat thoughtful limitations section.

**Reasons To Reject:**

- This is more of a question than a weakness. The explanations in Line 097-099 is somewhat weak. Would this bias some sentences (e.g., the first K sentences in a news article) that are designed to be summaries?

**Reproducibility:**

4: Could mostly reproduce the results, but there may be some variation because of sample variance or minor variations in their interpretation of the protocol or method.

**Reviewer Confidence:**

3: Pretty sure, but there's a chance I missed something. Although I have a good feel for this area in general, I did not carefully check the paper's details, e.g., the math, experimental design, or novelty.

---

> ### Author Rebuttal · Authors · 2023-08-28
>
> > This is more of a question than a weakness. The explanations in Line 097-099 is somewhat weak. Would this bias some sentences (e.g., the first K sentences in a news article) that are designed to be summaries?
>
> You raise a good point that some sentences can be more important than others, especially in news articles.  In practice, however, this turns out not to be a problem for our method.  For example, in Experiment 1 we have 4 datasets that are comprised of news articles (FakeNews, 20News, NYT, and BBCNews).  For each of these datasets, using the DocSplit method results in SOTA accuracy and F1 scores.
>
> We can think of several theoretical explanations for why this might be the case:
>
> 1. The random split ensures that "important" sentences are likely to end up in both positive instances, especially as the number of "important" sentences increases.
>
> 2. Existing research on large document embeddings suggests that the first sentences of a document are not really that much more important than other sentences.  If they were, then the standard BERT model with a small context window would perform just as well as models with a much larger context window (like LongFormer or BigBird).  But the wide range of research on these models summarized in Section 3.2 suggests that this is not the case.

---

### Official Review · Reviewer_7Pji · 2023-08-03

**Soundness:** 3

**Excitement:**

4: Strong: This paper deepens the understanding of some phenomenon or lowers the barriers to an existing research direction.

**Paper Topic And Main Contributions:**

This paper proposes a new contrastive pretraining method called DOCSPLIT which forces models to learn representation of documents with positive examples that are randomly sampled from the input document, and negative examples are randomly from unrelated documents. The proposed model achieves SOTA performance on document classification, few-shot learning, and retrieval tasks.

**Questions For The Authors:**

The random sampling method for positive instances generation is quite interesting. Is there exists the situation where the extracted summary may contain sentences that are not very relevant to the topic of the summary? I feel a bit confused about how do the authors make sure that the model is capable of generating high-quality positive instances with random sampling. Do the authors consider extracting sentences according to their contributions to the topic of the documents, e.g., attention weights?

**Reasons To Accept:**

The paper focuses on an interesting research topic and has clear research meaning. The scientific innovation is appropriate. The experiments are reasonable and convincing.

**Reasons To Reject:**

N/A

**Reproducibility:**

3: Could reproduce the results with some difficulty. The settings of parameters are underspecified or subjectively determined; the training/evaluation data are not widely available.

**Reviewer Confidence:**

3: Pretty sure, but there's a chance I missed something. Although I have a good feel for this area in general, I did not carefully check the paper's details, e.g., the math, experimental design, or novelty.

---

> ### Author Rebuttal · Authors · 2023-08-28
>
> These are great questions that we unfortunately weren't able to discuss in detail due to the short 4-page limitation, but we'll try to answer for you here.
>
> > Is there exists the situation where the extracted summary may contain sentences that are not very relevant to the topic of the summary? I feel a bit confused about how do the authors make sure that the model is capable of generating high-quality positive instances with random sampling.
>
> Random sampling is a standard component in most methods that use contrastive learning.  For example, our baseline models (CT-BERT, SimCSE, Contriever, and INSTRUCTOR) all use random sampling to some degree in their loss functions.
>
> Intuitively, our use of randomness is similar to the randomness in stochastic gradient descent.  We can't "make sure" that all of the samples are high quality because of the randomness, but the goal is that the low quality samples are relatively few and will and so the training will still converge to a good solution.  In Figure 1, we provide one example random split of a document, but if you try any of the other possible splits you would see a similarly reasonable result.
>
> > Do the authors consider extracting sentences according to their contributions to the topic of the documents, e.g., attention weights?
>
> This is an interesting idea that we've not thought about too much.  Our initial reaction is that it is unlikely to work well for several reasons:
>
> 1. Random splitting is much faster than another model would be.  Using another model to select the division would result in approximately a 2x slowdown in training speed, which would mean that the training dataset would be about half the size for a given computational budget.  Since larger datasets have been shown to have such an important effect on performance for modern NLP models, the alternative model would have to be quite good indeend to justify using half the amount of data.
>
> 2. There's no "obvious" way to use another model to perform the splitting.  It's reasonable to try to exploit the attention weights, but since these are at a token level rather than the sentence level, we would need to somehow aggregate these weights.  The obvious aggregation choices are ave/max pooling, but this results in a vector output, and there's no obvious way for converting that vector into a binary splitting decision.

---

### Official Review · Reviewer_yVuF · 2023-08-03

**Typos Grammar Style And Presentation Improvements:** 1. In 3.2, the attention is presented…
**Soundness:** 3

**Excitement:**

3: Ambivalent: It has merits (e.g., it reports state-of-the-art results, the idea is nice), but there are key weaknesses (e.g., it describes incremental work), and it can significantly benefit from another round of revision. However, I won't object to accepting it if my co-reviewers champion it.

**Paper Topic And Main Contributions:**

This paper proposes an unsupervised pre-training objective to improve document embeddings for text classification and document retrieval. A contrastive loss is designed to distinguish document sections of the same documents from those of different documents, which is optimized together with the masked language model loss. The method is shown to be effective in both short and long document settings against several baselines.

**Questions For The Authors:**

1. Have you tried to train the baseline models (only MLM loss) using your data and hyperparameters? Using only English wiki may not always be a disadvantage if the downstream tasks match the domain, and the hyperparameters in pre-training also affects the results significantly as shown by the roberta paper.

**Reasons To Accept:**

1. The proposed method is simple and model-agnostic.
2. The method works for both short and long documents.

**Reasons To Reject:**

1. The work is incremental, and lacks novelties.


**Reproducibility:**

4: Could mostly reproduce the results, but there may be some variation because of sample variance or minor variations in their interpretation of the protocol or method.

**Reviewer Confidence:**

5: Positive that my evaluation is correct. I read the paper very carefully and I am very familiar with related work.

---

> ### Author Rebuttal · Authors · 2023-08-28
>
> > Have you tried to train the baseline models (only MLM loss) using your data and hyperparameters?  Using only English wiki may not always be a disadvantage if the downstream tasks match the domain, and the hyperparameters in pre-training also affects the results significantly as shown by the roberta paper.
>
> You have a valid concern about datasets and hyperparameters effecting performance on downstream tasks.  We did not perform the experiment you suggest, however, because the SimCSE paper already performed this experiment and our experimental design is modeled on theirs.
>
> In particular, the SimCSE paper trained models on only English-language wikipedia, including retraining the BERT model with the MLM loss on this dataset.  They then evaluated on 7 semantic text similarity (STS) downstream tasks and found that the MLM-loss BERT model performed significantly worse than than the standard BERT checkpoint for these tasks.  This suggests that the reduced dataset size of using only Wikipedia has a negative impact on the BERT model's performance for downstream tasks, and that the standard checkpoint should be used instead for best performance.
>
> We also note that follow-on papers with similar experimental setups (e.g. CT-BERT and Contriever) compare only against the BERT checkpoint and not against a retrained-from-scratch BERT.
>
> Based on this explanation, we hope that you will consider increasing your soundness evaluation of our paper.  We did not include a discussion like this in the original submission due to space limitations, but we will try to include this discussion in the camera-ready copy.
>
> > In 3.2, the attention is presented as O(n^2) time complexity, which is only true for the vanilla implementation but not for attention mechanism itself. (see https://arxiv.org/abs/2112.05682)
>
> We believe this is incorrect.  In particular, the paper you linked provides algorithms for improving the *memory* complexity, but not the *time* complexity.  For example, the paper abstract states that "the time complexity is still O(n^2)".  We believe that the only known mechanisms for improving the time complexity of attention remain approximations such as the LongFormer, BigBird and other models cited in Section 3.2.
>
> This is still a valuable citation, however, that we are grateful for, and we will add it to our discussion in Section 3.2.  One of the main advantages of our DocSplit model is that it is entirely model agnostic, and so will work with the memory efficient algorithms proposed by this paper, and that is something that we would like to highlight.

---

### Meta-Review · Area_Chair_r8Hu · 2023-09-18

**Recommendation:** 2

**Metareview:**

This paper proposes a simple idea that improves learning of document embeddings. The idea is to apply standard a standard contrastive learning technique (operating at the below-document section level) that instills 'global' (i.e., document-level) understanding which in turn then enhances learning document embeddings. Contrastive learning operates, as usual, on the sets of positive and negative pairs, where the positive instances are sections sampled from the same document and negative instances are samples randomly taken from unrelated documents. Given that this is a short paper, it does a good coverage of tasks related to learning good document embeddings and it shows some improvements over standard document embedding models (e.g., Longformer, BigBird, Contriever). As mentioned (and also clearly stated by the authors), the idea is very simple, but it yields consistent gains in the three evaluation tasks.

I agree with one of the reviewers that the paper should have ideally investigated the idea of contrastive learning also with non-random (e.g., topically more similar) negative samples which resembles the idea of using hard negatives in standard contrastive learning frameworks. The paper also does not disclose which exact contrastive objective/loss is used (as there are plenty of different options on how to implement the basic contrastive idea, and it is also possible to combine different losses).

However, my main and strong concern is that, although the gains over the chosen baselines are salient, the paper does not provide a good comparison to all the relevant baselines and does not cover those recent methods and baselines in future work. The lack of these comparisons reduces the potential impact and it might negatively affect the novelty of the proposed method (again, noting its sheer simplicity). For instance, a body of work aims to improve (long) document embeddings by running specific contrastive learning techniques and the authors should ideally acknowledge and compare against the following papers at least:
- https://arxiv.org/abs/2305.16031
- https://aclanthology.org/2022.emnlp-main.802
- https://arxiv.org/abs/2103.14542
- https://aclanthology.org/2021.emnlp-main.109
- https://aclanthology.org/2021.findings-emnlp.327.pdf

---

### Decision · Program_Chairs · 2023-10-07

**Decision:**

Accept-Findings

**Comment:**

This paper proposes a simple idea that improves learning of document embeddings. The idea is to apply standard a standard contrastive learning technique (operating at the below-document section level) that instills 'global' (i.e., document-level) understanding which in turn then enhances learning document embeddings. Contrastive learning operates, as usual, on the sets of positive and negative pairs, where the positive instances are sections sampled from the same document and negative instances are samples randomly taken from unrelated documents. Given that this is a short paper, it does a good coverage of tasks related to learning good document embeddings and it shows some improvements over standard document embedding models (e.g., Longformer, BigBird, Contriever). As mentioned (and also clearly stated by the authors), the idea is very simple, but it yields consistent gains in the three evaluation tasks.

I agree with one of the reviewers that the paper should have ideally investigated the idea of contrastive learning also with non-random (e.g., topically more similar) negative samples which resembles the idea of using hard negatives in standard contrastive learning frameworks. The paper also does not disclose which exact contrastive objective/loss is used (as there are plenty of different options on how to implement the basic contrastive idea, and it is also possible to combine different losses).

However, my main and strong concern is that, although the gains over the chosen baselines are salient, the paper does not provide a good comparison to all the relevant baselines and does not cover those recent methods and baselines in future work. The lack of these comparisons reduces the potential impact and it might negatively affect the novelty of the proposed method (again, noting its sheer simplicity). For instance, a body of work aims to improve (long) document embeddings by running specific contrastive learning techniques and the authors should ideally acknowledge and compare against the following papers at least:
- https://arxiv.org/abs/2305.16031
- https://aclanthology.org/2022.emnlp-main.802
- https://arxiv.org/abs/2103.14542
- https://aclanthology.org/2021.emnlp-main.109
- https://aclanthology.org/2021.findings-emnlp.327.pdf